# Resource availability enhances positive tree functional diversity effects on carbon and nitrogen accrual in natural forests

Xinli Chen [1,2,3] ✉, Peter B. Reich[3,4,5], Anthony R. Taylor [6], Zhengfeng An[2] & Scott X. Chang [1,2] ✉

Forests harbor extensive biodiversity and act as a strong global carbon and nitrogen sink. Although enhancing tree diversity has been shown to mitigate climate change by sequestering more carbon and nitrogen in biomass and soils in manipulative experiments, it is still unknown how varying environmental gradients, such as gradients in resource availability, mediate the effects of tree diversity on carbon and nitrogen accrual in natural forests. Here, we use Canada's National Forest Inventory data to explore how the relationships between tree diversity and the accumulation of carbon and nitrogen in tree biomass and soils vary with resource availability and environmental stressors in natural forests. We find that the positive relationship between tree functional diversity (rather than species richness) and the accumulation of carbon in tree biomass strengthens with increasing light and soil nutrient availability. Moreover, the positive relationship between tree functional diversity and the accumulation of carbon and nitrogen in both organic and mineral soil horizons is more pronounced at sites with greater water and nutrient availabilities. Our results highlight that conserving and promoting functionally diverse forests in resource-rich environments could play a greater role than in resource-poor environments in enhancing carbon and nitrogen sequestration in Canada's forests.

Forests play a pivotal role in the exchange of carbon (C) between the biosphere and atmosphere, sequestering about 30% of annual global anthropogenic $CO_2$ emissions[1]. In addition, nitrogen (N) availability in forest soils is critical in determining plant growth and associated C sequestration[2]. Consequently, enhancing the capacity of forests to sequester more C in plants and soils while retaining greater soil N to support C sequestration is an important nature-based climate solution to help limit global warming to 1.5 °C by the end of this century[3]. Recent studies have emphasized that increasing tree diversity is a powerful nature-based strategy for enhancing C sequestration[4], as forest productivity and soil C and N accumulation are often higher in species-rich tree communities over monocultures[5–8]. However, available evidence, especially for plant diversity effects on soil C and N accumulation, primarily comes from designed biodiversity manipulation experiments conducted in homogeneous conditions[6,8]. It remains unclear how positive biodiversity-ecosystem functioning (BEF) relationships vary with resource availability (e.g., light, water, and soil nutrients) and background environmental conditions in heterogeneous natural forests[9–11]. Understanding the role of resource availability and other background environmental factors in shaping BEF relationships is necessary to improve predictive BEF models and inform forest management and conservation.

[1]State Key Laboratory of Subtropical Silviculture, Zhejiang A&F University, Hangzhou, China. [2]Department of Renewable Resources, University of Alberta, Edmonton, AB, Canada. [3]Institute for Global Change Biology, and School for Environment and Sustainability, University of Michigan, Ann Arbor, MI, USA. [4]Department of Forest Resources, University of Minnesota, St. Paul, MN, USA. [5]Hawkesbury Institute for the Environment, Western Sydney University, Penrith, NSW, Australia. [6]Faculty of Forestry and Environmental Management, University of New Brunswick, Fredericton, NB, Canada. ✉e-mail: xinlichen@zafu.edu.cn; sxchang@ualberta.ca

The role of 'facilitation' and 'resource (niche) partitioning'—two major mechanisms that contribute to efficient ecosystem functioning in species-rich plant communities— might shift with changes in resource availability and non-resource-related environmental stressors (hereafter referred to as 'non-resource stressors')[12] (Fig. 1). Specifically, facilitation (considered a form of complementarity) occurs when some species in a plant assemblage enhance resource availability and/or modify the environment in such a way that it alleviates stress and enhances productivity of other species, and thus for the entire plant community[12]. Based on the 'stress-gradient hypothesis,' the role of facilitation may become stronger as environmental stresses increase through stress amelioration[13] (Fig. 1). For example, based on the Swiss National Forest Inventory data, both spruce and fir have been found to be more productive in mixtures than in monocultures at sites experiencing high acidity stress, rather than those with optimum soil pH[14]. However, the generality of the stress-gradient hypothesis is contentious, particularly in its applicability to gradients in resource availability, as it ignores the high dependence of facilitation on the presence of resource-enrichment facilitative species[10,15]. In the absence of such species, resource competition would come to dominate plant-plant interactions when resources are more limited[15,16]. Alternatively, the resource partitioning effect (another form of complementarity) occurs when coexisting species in a plant assemblage employ distinct strategies to harvest resources by partitioning their acquisition in space, time, or chemical form[12]. Compared with facilitation, resource partitioning would consistently work to enhance ecosystem functioning[12], and might show a hump-shaped pattern as resource availability increases. In extremely resource-limited environments, the role of resource partitioning is less pronounced due to the scarcity of resources available for partitioning. As resource limitations become moderate, resource partitioning likely plays a more significant role due to the greater availability of resources for diverse plant communities to use in resource partitioning[10,15]. However, in environments rich in resources, plant communities are less dependent on resource partitioning to increase productivity, as resources themselves are no longer a limiting factor[12]. In natural high-latitude forests, typically constrained by multiple resources, we expect that positive BEF relationships would be strengthened with increasing resource availability (Fig. 1).

Manipulative experiments have revealed that the positive effects of plant diversity on productivity were stronger under resource enrichment, such as the addition of water and N, and higher levels of temperature, light, and $CO_2$ availability[17–20]; however, these experiments are constrained by the size of the experimental unit and scope, thus limiting the inference space for projecting large-scale patterns. Recently, large-scale observational studies of natural forests have been employed to explore the relationship between tree diversity and productivity along climate gradients and equivocal results have been reported[21–24]. Some studies are aligned with the stress gradient hypothesis, showing a more pronounced positive association between tree diversity and productivity in drier than in more humid regions[21,22], while others reveal increasing positive relationships with rising water availability[23,24]. Moreover, relationships between tree diversity and C and N accumulation on a large scale have rarely been examined within the framework of both resource availability and non-resource stress gradients, especially for soil C and N accumulation. Previous large-scale observational studies generally evaluated relationships between tree diversity and soil C and N stocks at one point in time, possibly overlooking the enduring influence of historical plant compositions, which can shape soil C and N storage for centuries or even millennia[25]. Compared with soil C and N stocks that reflect long-term net accumulation over centuries, soil C and N accumulation during recent decades should better reflect the effects of recent tree diversity and provide a standardized timeframe for comparison[7]. Finally, although temperature is not traditionally considered a resource (as it is not consumable), it often serves as a surrogate for chemical energy availability in addition to light since higher temperatures are generally more favorable for productivity and soil C and N cycling in high-latitude forests as long as there is sufficient water available as well[26–28].

In this study, we investigate how resource availability and non-resource stressors influence the relationship between tree diversity and the accumulation of C and N in trees and forest soils (Supplementary Figs. 1–3). We analyze the decadal accumulation of C in tree biomass, and C and N in the organic and mineral soil horizons as a function of tree diversity and their interaction with light, water and N availability, air temperature, and non-resource stressors (i.e., heatwaves and low soil pH) gradients while controlling for the influence of forest stand age and tree functional identity. Diversity in functional traits (hereafter referred to as 'functional diversity', $FD_{is}$) within each inventory plot tree community is used to represent tree diversity, as it better captures resource partitioning effects[29]. In this paper, we specifically hypothesize that the positive relationship between tree functional diversity and the

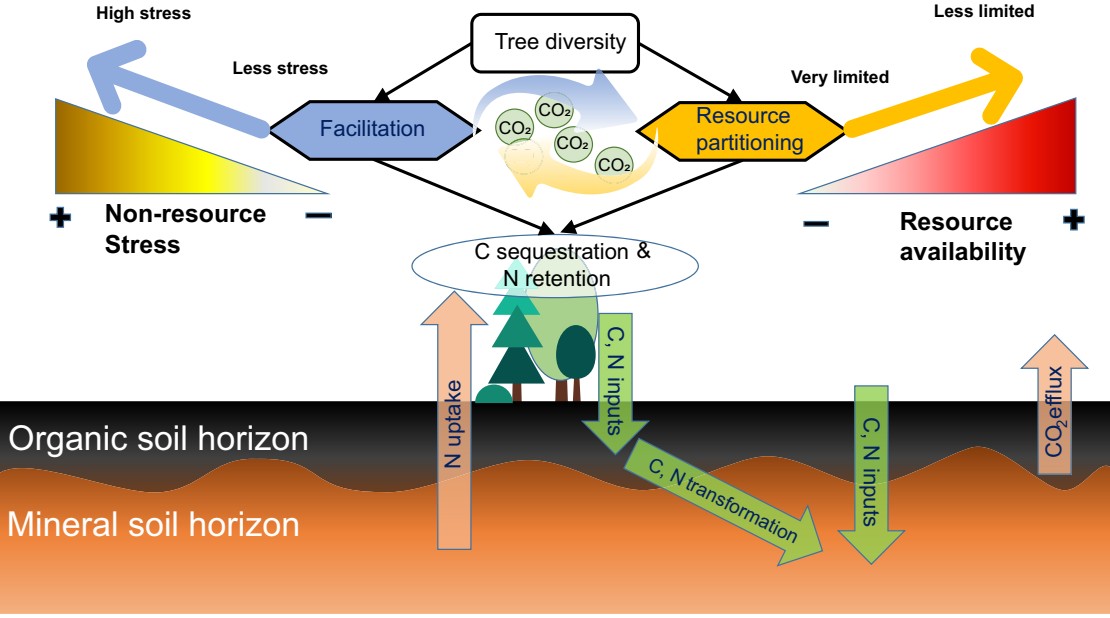

**Fig. 1 | A conceptual diagram of environmental context-dependent effects of tree diversity on C and N accumulation in tree biomass and forest soils.** Facilitation and resource partitioning are two primary mechanisms that drive efficient ecosystem functioning in species-rich plant communities.

accumulation of C in tree biomass and C and N in the soil will be stronger in resource (light, water, and N)-rich environments and areas experiencing higher heatwave or acidity stress, due to enhanced resource partitioning and facilitation, respectively. When assessing how relationships between tree functional diversity and changes in C and N accumulation vary with resource availability, we simultaneously consider three resource factors—light (mean annual solar radiation, hereafter solar radiation), water (mean annual climate moisture index, hereafter CMI), and nitrogen (decadal cumulative N deposition rate, hereafter N deposition) availability—along with their interactions with $FD_{is}$. In addition, soil C/N ratios in both the organic and mineral soil horizons are used as a surrogate of site nutrient status to replace N deposition in the full models. When assessing how the relationship between tree functional diversity and changes in C and N accumulation vary with non-resource stress variables, we conduct separate analyses for mean annual heatwave intensity (hereafter HI) and soil pH, given the independence of their effects on soil C and N stocks and the lack of soil pH data from some sites. Given the high geographical intercorrelation between solar radiation and MAT ($R^2 = 0.34$, Supplementary Fig. 4), we take a stepwise approach to disentangling their impacts on C and N accumulation (see details in Methods). To prevent overfitting, the most parsimonious models are selected based on the lowest Akaike information criterion (AIC) (Fig. 2). These analyses are based on data from the first (2000–2006) and second (2008–2017) census of the Canadian National Forest Inventory (NFI) permanent sample plot network, which offers a broad representation of Canada's temperate and boreal forests.

## Results

We found that interactions between tree functional diversity and the environmental drivers affected C accumulation in trees and the accumulation of both C and N in the soil (Supplementary Tables 1–4). Solar radiation and soil C/N were the main resource variables that modulated relationships between $FD_{is}$ and tree C accumulation in the most parsimonious (with the lowest AIC) models. In contrast, CMI, N deposition, and soil C/N were the main variables that modulated C and N accumulation in organic and mineral soil horizons (Fig. 2). The HI was included as the main non-resource stress explanatory variable for all C and N accumulation variables except the decadal soil C change in the mineral horizon. In comparison, soil pH was only retained as the main stress predictor for C accumulation in tree biomass in the most parsimonious model (Fig. 2).

## Resource availability and tree diversity affect tree biomass C accumulation

With respect to trees, solar radiation was positively related to tree biomass C accumulation in the most parsimonious model (F = 22.45, degrees of freedom (df) = 1, P = 0.000003) (Supplementary Table 1, Fig. 3a). Tree biomass C accumulation was also positively associated with MAT (F = 32.93, df = 1, $P = 1.6 \times 10^{-8}$) (Supplementary Fig. 5a). However, we found that solar radiation was still positively correlated with C accumulation in tree biomass when modeling the residuals from the regression of tree biomass C accumulation on MAT as a function of solar radiation and its interaction with $FD_{is}$ (F = 6.64, df = 1, P = 0.010, Supplementary Fig. 5b), indicating that C accumulation in tree biomass significantly increases with solar radiation, even after accounting for the effects of MAT. In addition, C accumulation in tree biomass was inversely correlated with the organic horizon soil C/N (F = 14.91, df = 1, P = 0.0001), and pH (F = 6.47, df = 1, P = 0.012) (Fig. 3c, e).

The accumulation of C in tree biomass was positively correlated with $FD_{is}$, and this relationship was dependent on environmental variables (Supplementary Table 1, Fig. 3). In line with our hypothesis, the increase in tree biomass C accumulation with $FD_{is}$ was more pronounced at sites with high solar radiation and low soil C/N (i.e., relatively N-rich soils), compared to sites with low solar radiation (F = 7.54, df = 1, P = 0.006, Fig. 3b) and high soil C/N (i.e., relatively N-poor soils) in the organic horizon (F = 8.30, df = 1, P = 0.004, Fig. 3d). In addition, the positive association between tree $FD_{is}$ and the C accumulation in tree biomass marginally shifted from neutral at sites with optimum soil pH (pH = 7.0) to positive at sites with high acidity (pH = 4) (F = 2.88, df = 1, P = 0.091, Fig. 3f).

## Resource availability and tree diversity affect soil C and N accumulation

We found that C accumulation in the organic soil horizon did not increase with tree $FD_{is}$ (F = 2.99, df = 1, P = 0.085), but that the accumulation of C in the mineral soil horizon was positively related to tree $FD_{is}$ (F = 6.69, df = 1, P = 0.010) (Supplementary Tables 2, 3). However, the associations between $FD_{is}$ and the accumulation in soil C were conditional on at least one of several environmental variables (CMI, N deposition, or HI) (Supplementary Tables 2, 3). The association between $FD_{is}$ and C accumulation in the organic horizon shifted from negative at low-CMI sites to positive at high-CMI sites (F = 13.20, df = 1, P = 0.0003, Fig. 4a). In addition, the association between $FD_{is}$ and C accumulation in

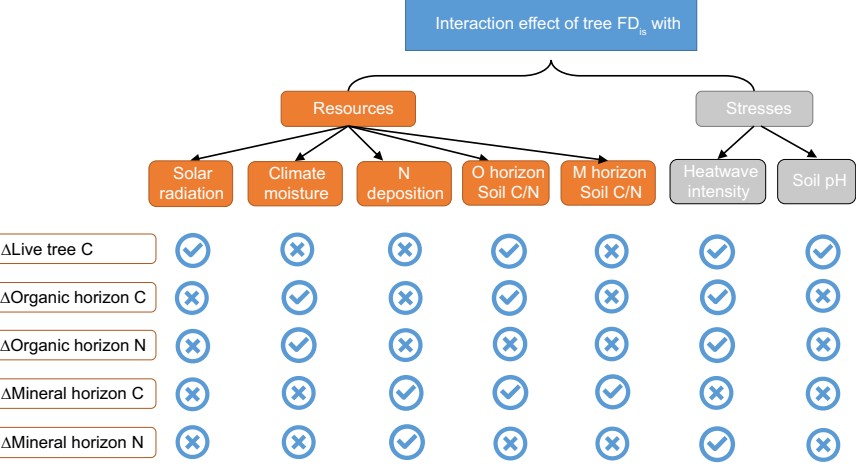

**Fig. 2 | A simplified diagram showing the included interaction effects of tree functional diversity (FDis) and environmental factors on C accumulation in tree biomass and both C and N accumulation in the organic (O horizon) and mineral (M horizon) soil horizons in the most parsimonious model.** The symbols √ and × indicate whether the interaction between tree functional diversity and selected variables was included in the most parsimonious model. The Δ symbol indicates changes over time.

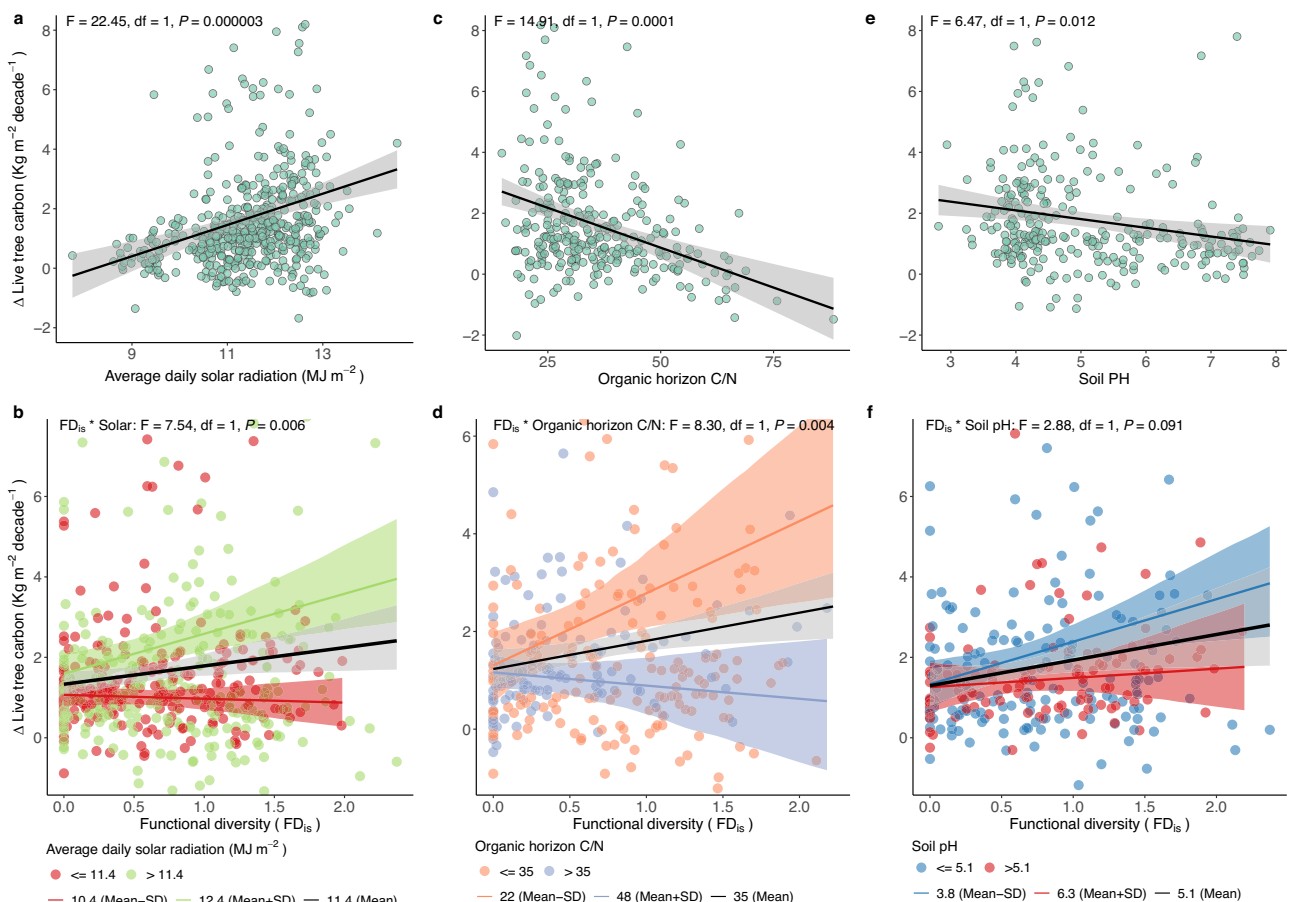

**Fig. 3 | Environmental context-dependent responses of tree biomass C accumulation to functional diversity (FDis). a, b** Solar-dependent responses for residual tree biomass C accumulation (as indicated by the Δ symbol) after removing MAT effect; **c, d** organic horizon soil C/N-dependent responses; **e, f** soil pH-dependent responses. The black line and grey shaded areas in panels **a, c**, and e represent the fitted regression and its bootstrapped 95% confidence intervals. Colored lines in panels **b, d**, and **f** represent the Solar-, soil C/N-, and pH-specific responses, with their bootstrapped 95% confidence intervals shaded in the corresponding color. Solar, soil C/N, and soil pH were analyzed as continuous variables but illustrated here based on the meaningful three levels of breakpoints (mean, mean plus, and minus standard deviation (SD)). The significance (*P*) is reported for each term tested, with *P* values calculated using a one-sided F test. Solar: decadal average of annual solar radiation (for each plot), df: degrees of freedom.

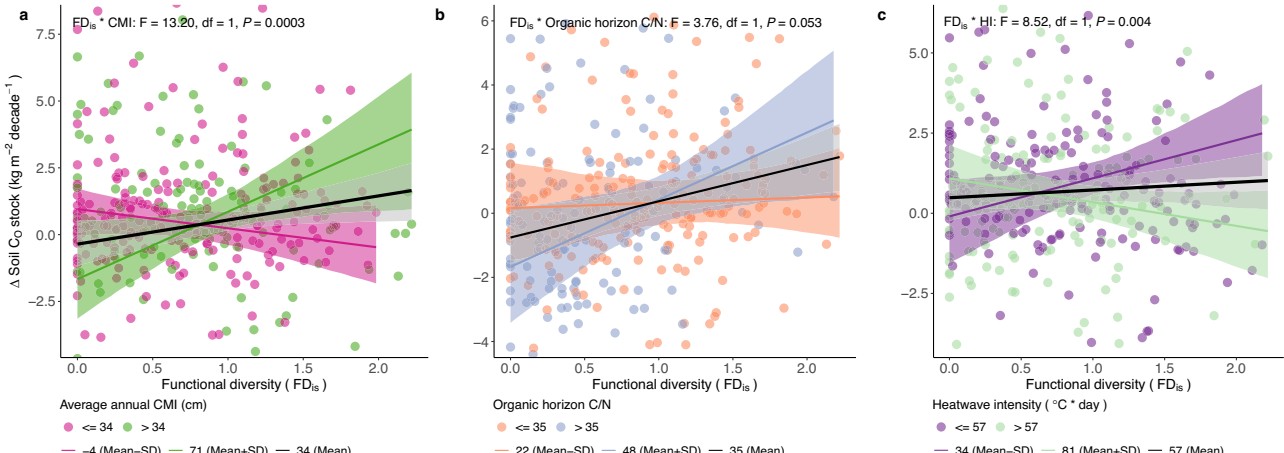

**Fig. 4 | Environmental context-dependent responses of soil C accumulation in the organic horizon to functional diversity (FDis). a** CMI-dependent responses; **b** organic horizon soil C/N-dependent responses; **c** HI-dependent responses. Colored lines represent the CMI-, soil C/N-, and HI-specific responses, with their bootstrapped 95% confidence intervals shaded in the corresponding color. CMI, soil C/N, and HI were analyzed as continuous variables but illustrated here based on the meaningful three levels of breakpoints (mean, mean plus, and minus standard deviation (SD)). The significance (*P*) is reported for each term tested, with *P* values calculated using a one-sided F test. CMI: decadal average of annual climate moisture index (for each plot), HI: decadal average of annual heatwave intensity (for each plot). df: degrees of freedom.

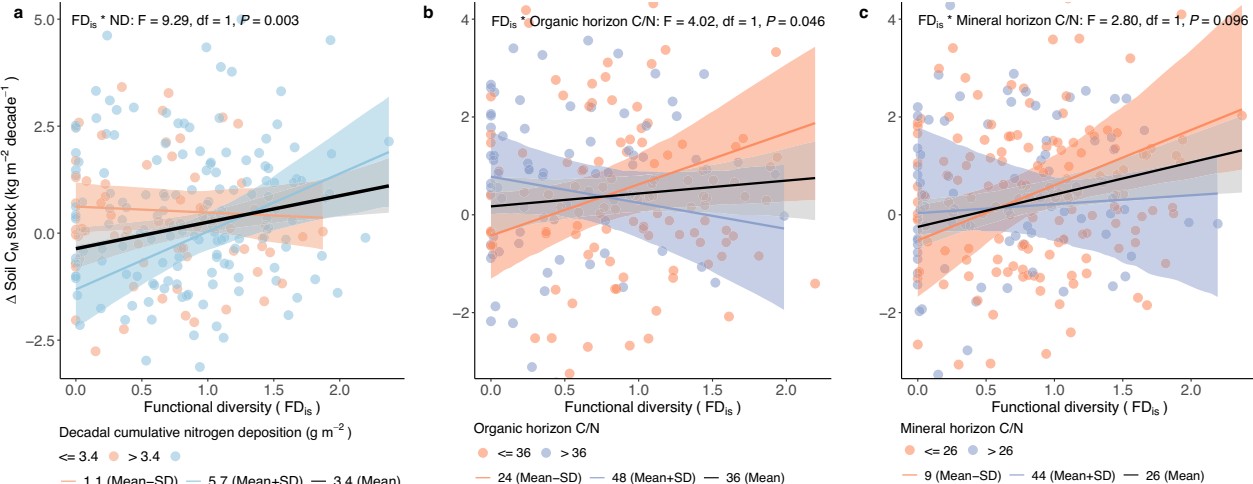

**Fig. 5 | Environmental context-dependent responses of soil C accumulation in the mineral horizon to functional diversity (FD$_{is}$). a** N deposition-dependent responses; **b** organic horizon soil C/N-dependent responses; **c** mineral horizon soil C/N-dependent responses. Colored lines represent the N deposition- and soil C/N-specific responses, with their bootstrapped 95% confidence intervals shaded in the corresponding color. N deposition and soil C/N were analyzed as continuous variables but illustrated here based on the meaningful three levels of breakpoints (mean, mean plus and minus standard deviation (SD)). The significance ($P$) is reported for each term tested, with $P$ values calculated using a one-sided F test. ND: decadal cumulative of N deposition (for each plot), df: degrees of freedom.

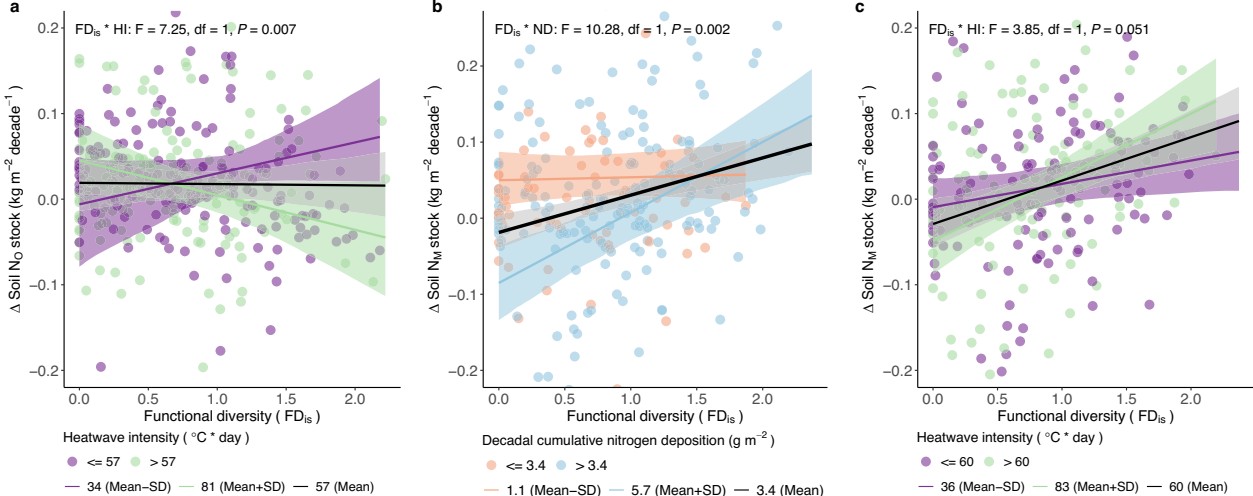

**Fig. 6 | Environmental context-dependent responses of soil N accumulation in the organic and mineral horizons to functional diversity (FD$_{is}$). a** CMI-dependent N accumulation in the organic horizon; **b** N deposition-dependent N accumulation in the mineral horizon; **c** HI-dependent N accumulation in the mineral horizon. Colored lines represent the CMI, N deposition, and HI-specific responses, with their bootstrapped 95% confidence intervals shaded in the corresponding color. CMI, N deposition, and HI were analyzed as continuous variables but were illustrated here based on meaningful three levels of breakpoints (mean, mean plus and minus standard deviation (SD)). The significance ($P$) is reported for each term tested, with $P$ values calculated using a one-sided F test. CMI: decadal average of annual climate moisture index (for each plot), ND: decadal cumulative N deposition (for each plot), HI: decadal average of annual heatwave intensity (for each plot), df: degrees of freedom.

the organic horizon marginally shifted from neutral at sites with low soil C/N to positive at sites with high soil C/N in the organic horizon (F = 3.76, df = 1, $P$ = 0.053, Fig. 4b). In addition, the C accumulation in the organic horizon was positively correlated with FD$_{is}$ at sites with low HI, but negatively correlated with FD$_{is}$ at sites with high HI (F = 8.52, df = 1, $P$ = 0.004, Fig. 4c). Moreover, the positive association between FD$_{is}$ and C accumulation in the mineral horizon was (significantly or marginally significantly) more pronounced at N-rich sites that were characterized by high N deposition (F = 9.29, df = 1, $P$ = 0.003) and low soil C/N in both the organic and mineral horizons (F = 4.02, df = 1, $P$ = 0.046 and F = 2.80, df = 1, $P$ = 0.096, respectively) (Fig. 5).

The N accumulation in the organic soil horizon did not increase with tree FD$_{is}$ ($F$ = 0.99, df = 1, $P$ = 0.321), but that in the mineral soil horizon was positively related to tree FD$_{is}$ ($F$ = 11.66, df = 1, $P$ = 0.0008) (Supplementary Table 4). Similar to C accumulation, the N accumulation in the organic and mineral horizon were, respectively, positively correlated with FD$_{is}$ at sites with low HI ($F$ = 7.25, df = 1, $P$ = 0.007) and high N deposition ($F$ = 10.28, df = 1, $P$ = 0.002), rather than sites with high HI and low N deposition (Fig. 6a, b). In contrast, the relationship between FD$_{is}$ and N accumulation in the mineral soil horizon marginally shifted from neutral at sites with low HI to positive at sites with high HI ($F$ = 3.85, df = 1, $P$ = 0.051) (Fig. 6c).

In addition, we also tested the effects of species richness and evenness on C and N accumulation by replacing the terms of FD$_{is}$ in the most parsimonious model with species richness and evenness. Our results revealed significant interactive effects of species evenness and

several environmental factors (solar radiation, CMI, N deposition, and HI) on C and N accumulation in tree biomass and soils, largely mirroring those of significant interactive effects of functional diversity and environmental factors (Supplementary Tables 1–5). In contrast, the interactive effects of species richness with environmental variables on C and N accumulation in trees and soils were generally non-significant (Supplementary Table 6), suggesting that equitable distribution of functional traits, instead of the number of species, drives changes in C and N sequestration in natural forests across resource and non-resource gradients.

## Discussion

This study shows that the relationship between $FD_{is}$ and the decadal accumulation of C and N in tree biomass and forest soils is dependent on resource availability and non-resource stress variables, underscoring the importance of considering the environmental context in studying biodiversity-ecosystem functioning relationships. Specifically, we found that the positive association between tree functional diversity and C accumulation in tree biomass was more pronounced in solar radiation-, and nutrient-rich environments. Additionally, the positive relationship between functional diversity and C and N accumulation in the organic soil horizon was more pronounced in high-water and low-heatwave conditions, while in mineral soil horizons, these relationships were stronger in nutrient-rich and high-heatwave environments. Our results validate the results from small-scale experimental studies that manipulate plant diversity, resource availability, and environmental conditions[17,18].

The increased C accumulation in tree biomass with increasing solar radiation conditions is consistent with previous studies[26,30], suggesting that the productivity of Canada's forests is a function of temperature, precipitation, and solar radiation. Given that solar radiation was significantly correlated with MAT in cold forests (MAT < 3 °C), but not in warmer forests (MAT ≥ 3 °C) (Supplementary Fig. 4), the lower C accumulation in tree biomass that we observed in regions with low solar radiation may result not only from limited energy for photosynthesis but also from lower MAT[31]. Thus, the positive relationship between tree C accumulation and solar radiation observed here should be interpreted cautiously, especially in cold regions. Regarding non-resource stress factors, prolonged extreme warmth would result in a decline of net photosynthesis, as indicated by the lower rate of tree C accumulation observed under greater heatwave intensity[32]. Despite the expected aluminum toxicity in low acidic soils (typically seen in soils with a calcium/aluminum ratio <1), we observed greater tree C accumulation in highly acidic soils (pH around 2.8) than in optimum soils (pH around 7.0), which collaborates with results from a US national forest inventory-based study[33]. The absence of calcium deficiency in boreal soils, unlike tropical soils may partly explain why trees in highly acidic soils still exhibit robust growth[34]. In addition, decreasing soil pH could enhance the availability of some soil nutrients (e.g., copper and iron), and promote the establishment of ectomycorrhizal fungal-associated tree species, resulting in an increase in forest productivity[33,35].

Importantly, our results show that the positive relationship between tree functional diversity and C accumulation in tree biomass appears to strengthen under higher solar radiation and N availability (characterized by low C/N), indicating that the resource partitioning effect in functionally diverse forests can benefit from increased resource availability[17,19]. At sites with relatively higher solar radiation and available N, the greater light and N capture capacity of more diverse forests[5,36] would use the light and N more completely, resulting in increased production of tree biomass[19].

Our results also showed that functionally diverse tree communities had greater C accumulation in the organic horizon in humid environments. Enhanced tree diversity may lead to increased litter input[37], thereby boosting C accumulation in the organic horizon at wetter sites,

where organic matter decomposes more slowly under wetter conditions[38]. In comparison, less diverse tree communities had greater C accumulation in the organic horizon in drier environments. Given that plant functional diversity could also increase decomposition rates in the organic horizon[37], our results suggest that in more humid environments, tree diversity might enhance productivity more than it accelerates decomposition and mineralization processes, whereas, in drier environments, tree diversity potentially enhances productivity less than it accelerates decomposition and mineralization processes, similar to findings in grasslands[39]. The wetter condition that slows organic matter decomposition discussed earlier may also counteract the enhancement of functional diversity on organic matter decomposition. Moreover, the more pronounced positive associations between tree functional diversity and C and N accumulation in the mineral horizon at sites with relatively high N availability indicate that increased soil N availability benefits resource partitioning in high- than low-diversity tree communities, enhancing root productivity and rhizodeposition[40,41], thereby resulting in higher C and N accumulation in the mineral horizon.

In contrast with our hypothesis, the positive relationship between tree functional diversity and C accumulation in the organic horizon was marginally stronger at low N sites characterized by organic matter with high C/N ratios. It is possible that high C/N ratios in the organic horizon slow down the decomposition rate and, in so doing, accumulate the C from enhanced litter input in diverse tree communities[42]. Given the antagonistic effects between water availability and nitrogen enrichment on the impact of tree functional diversity on organic horizon C accumulation, more studies are needed to examine the interaction effect of precipitation and N addition in regulating the relationship between tree diversity effects and soil C storage. Our results collectively suggest that a reduction in tree functional diversity might lead to greater C and N loss in forest biomass and mineral soils that are subject to increased nitrogen input in the future.

Regarding non-resource stress factors, we found a more pronounced positive relationship between functional diversity and tree C accumulation in highly acidic soils (though only marginally significant), while N accumulation in the mineral horizon increased with tree functional diversity in environments with high heatwave stress, partly validating the stress-gradient hypothesis[12]. However, functionally diverse tree communities had greater C and N accumulation in the organic horizon in environments with low heatwave stress, while less diverse tree communities (e.g., monocultures) showed greater C and N accumulation in the organic horizon under strong and prolonged heatwave stresses, probably because of the enhanced decomposition rate of the organic horizon under high heatwave stress[43]. In addition to being a stressor, heatwave events can provide energy to enhance temperature-dependent ecosystem processes[43], such as the decomposition of organic matter in the organic horizon, thus attenuating the increased litter input associated with higher tree functional diversity. We note that functional diversity in Canada's predominantly boreal and hemi-boreal forests, with a range of 0–2.4 and an average of 0.7, is much lower than those in tropical and subtropical forests[44]. Therefore, extrapolating our findings to ecosystems with higher functional diversity should be approached with caution. Since functional redundancy is also higher in tropical and subtropical forests[44], C sequestration in tree biomass might not change with functional diversity in ecosystems with a higher functional diversity[45]. Moreover, our models' predictive powers ($R^2$) are relatively low, which might be partly attributable to the relatively small sample sizes and the wide range of variability in non-measured climate, soil, disturbance regime and evolutionary history in conducting the measurements[46]. However, low predictive power at the individual sample level does not necessarily indicate the absence of significant relationships.

In conclusion, we show that relationships between tree diversity and ecosystem functioning are not constant across environmental gradients. Carbon accumulation in tree biomass might be substantially

increased by promoting tree functional diversity in areas with abundant light and nutrients. Similarly, enhancing tree functional diversity in regions with high water and nutrient availabilities might significantly boost soil C and N accumulation compared with regions with low water and nutrient availabilities. Moreover, we also highlight the stronger positive association (although only marginally significant) between tree functional diversity and tree C accumulation at sites with greater acidity stress, partly supporting the stress-gradient hypothesis. Given that N deposition might also result in soil acidification at the global scale in the future[47], our results suggest that reductions in tree diversity may reduce the capacity of forests to store C under N deposition and soil acidification. Therefore, efforts to promote tree diversity to enhance C sequestration and N retention for mitigating climate change and improving ecosystem sustainability must take into account the overarching role of local environmental conditions.

## Methods

### Study area and available data

We used plot-level data from the Canadian National Forest Inventory (NFI) database[48] to determine how tree diversity and environmental gradients interactively affect C and N accumulation in trees and soils. The NFI database encompasses a network of permanent ground plots covering much of Canada's forests across boreal and temperate biomes. The permanent plots were established and monitored by Canadian provincial authorities between 2000 and 2006 (first measurement) and subsequently re-measured between 2008 and 2017 (second measurement) following the same standard ground sampling guidelines established by the Canadian Forest Inventory Committee[49]. For inclusion in our analysis, we selected only those plots situated in forest stands that were unmanaged and had not been relocated at the time of sampling, with two measurements conducted and complete data coverage for forest canopy composition, stand age, tree C accumulation, soil C and N stocks. After excluding missing values for each horizon, 513 plots for canopy trees (Supplementary Fig. 1), 360 plots for organic soil horizon samples (Supplementary Fig. 2) and 244 plots for 0–15 cm mineral soil horizon samples (Supplementary Fig. 3) that span from 44°00′–64°24′ N to 53°24′–128°36′ W were included in the statistical analyses (Supplementary Table 7).

In each plot, which comprises several sub-plots, a 'Large Tree Plot' was established with a radius of 11.28 m and an area of 400 m$^2$ (0.04 ha). Within this Large Tree Plot, all canopy trees (tree stems ≥ 9.0 cm in diameter at breast height) were systematically numbered, tagged, identified for species, and measured for both height and DBH. The aboveground biomass of individual trees was estimated using published Canadian national species-specific DBH-based tree aboveground biomass allometric equations[50]. We then converted the aboveground biomass to the aboveground C stock of each tree by multiplying aboveground biomass with biome-level C concentrations of woody biomass based on Martin, Doraisami, and Thomas[51]. The decadal C accumulation in tree biomass due to tree growth (hereafter tree C accumulation) was calculated as the C increment by the growth of surviving trees and ingrowth by recruited trees between the two censuses[52]. In addition to the 'Large Tree Plot', four 1 m$^2$ "Microplots" were established outside of the large tree plot (but within a 15 m radius of the Large Tree Plot center). From each of these microplots, an organic horizon soil sample was collected that comprised the litter, fabric, and humus layers (over 17% organic C by mass) using 20 × 20 cm (inside dimensions) aluminum sampling frames[53]. Furthermore, seven mineral soil horizon samples (less than 17% organic C) were collected from each NFI microplot at fixed depths (0–15 cm) using a 10-cm diameter auger. The collected organic and mineral soil samples were dried at 70 °C in an oven and were subsequently sieved using 8 and 2 mm screens to eliminate gravel and roots, respectively. Soil C content, N content, pH, and bulk density were measured for each soil sample following standard protocols[49]. Soil C and N stocks were calculated by multiplying soil C and

N content by the soil bulk density of each soil horizon[54]. Then decadal changes in soil C and N stocks in the organic (ΔSoil $C_O$ stock and ΔSoil $N_O$ stock, kg m$^{-2}$ decade$^{-1}$) and mineral horizons (ΔSoil $C_M$ stock and ΔSoil $N_M$ stock, kg m$^{-2}$ decade$^{-1}$) were calculated as the difference in soil C and N stock between the two censuses divided by the census length in decades. We found that soil pH in the mineral horizon did not change with soil C/N ratios (Supplementary Fig. 6).

### Tree functional diversity and functional identity

To calculate functional diversity and identity, we employed five key functional traits that are linked to tree growth and resource acquisition abilities: 'leaf N content per leaf dry mass' ($N_{mass}$, mg g$^{-1}$), 'leaf phosphorus content per leaf dry mass' ($P_{mass}$, mg g$^{-1}$), 'specific leaf area' (SLA, mm$^2$ mg$^{-1}$; i.e., leaf area per leaf dry mass), 'wood density' (WD, g cm$^{-3}$), and maximum height (MH, m)[23,55–57]. We acquired the mean trait values for $N_{mass}$, $P_{mass}$, SLA, WD, and MH by aggregating all available measurements for each tree species from the TRY Plant Trait Database[58]. Functional dispersion was used as an index of tree functional diversity, which considers species relative abundances[59]. Functional dispersion was calculated as the average distance of individual tree species to the centroid in the multidimensional trait space of all tree species weighted by the relative basal area[59].

The stand age for each permanent sample plot was determined according to the date of the last stand-replacing fire or by coring three dominant/co-dominant trees in each plot. The stand age was represented by the mean stand age between the first and second NFI measurements.

The community-weighted means (CWMs) of the five traits, with weights reflecting basal area, were utilized to estimate the functional identity of the tree community using principal component analysis (PCA)[60–62]. Specifically, the first (CWM$_{PC1}$, explaining 47% of the variation) and the second axes (CWM$_{PC2}$, explaining 23% of the variation) of the PCA were used as proxies of functional identity. The CWM$_{PC1}$ represents traits associated with resource acquisitive abilities[55,63,64], and is strongly related to CWM$_{SLA}$, CWM$_{Nmass}$, and CWM$_{Pmass}$. The CWM$_{PC2}$ is related negatively to maximum height (CWM$_{MH}$) and positively to wood density (CWM$_{WD}$) (Supplementary Fig. 7). An assemblage of species characterized by fast growth rates (higher CWM$_{PC1}$) and large biomass stock (lower CWM$_{PC2}$) is expected to increase tree productivity and N retention and, therefore, soil C and N pools via increased tree litter input[65,66]. On the other hand, traits representing low resource acquisition abilities, such as low leaf N and P concentrations, specific leaf area, and high wood density, are also expected to contribute to soil C and N accumulation through the input of low-quality (recalcitrant) tree litter with slow decomposition rates[29,67]. The calculation of $FD_{is}$ and CWM was conducted using the FD package in R[59]. In order to account for the variation in tree diversity during these measurements, we used the mean values of $FD_{is}$ and CWM across the two censuses as proxies for tree diversity and identity[23]. The $FD_{is}$ slightly increased with solar radiation (F = 4.33, df = 1, P = 0.038) and N deposition (F = 6.35, df = 1, P = 0.012), but did not change with water availability (F = 0.42, df = 1, P = 0.516) (Supplementary Fig. 8).

### Local climate and soil condition

We used decadal mean annual solar radiation (solar radiation, MJ m$^{-2}$), mean annual climate moisture index (CMI, cm) and decadal cumulative N deposition (ND, g m$^{-2}$) preceding the second census as proxies for environmental resource availabilities. The solar radiation and CMI were extracted from the BioSIM software (https://cfs.nrcan.gc.ca/projects/133), which generates long-term, scale-free climate data based on geographic coordinates (latitude, longitude, and elevation)[68]. We used N deposition data at a 0.5° grid from ISIMIP[69,70]. The CMI was calculated as mean annual precipitation minus potential evapotranspiration[71]. The decadal mean annual heatwave intensity (HI) preceding the second census and mean soil pH between the two censuses were used as environmental non-resource stress indices. Higher HI values indicate

more intense heatwave events in the past ten years, while sites with low soil pH (acidity stress) might reduce plant growth by inhibiting the uptake of essential elements and causing specific-ion toxicities. The HI was defined as a period with 3 consecutive days or longer with daily mean temperatures greater than the 90th percentile of temperature over a 30-year historical baseline period[72]. The HI, calculated with the 'heatwaveR' package, represents an annual cumulative heatwave intensity value, where the intensity is determined by the difference between mean temperatures during each heatwave event and the threshold temperatures[72]. In addition to decadal cumulative N deposition, we also used the corresponding soil C/N ratios in each plot as an index of soil N condition. These plots were also grouped into two main biomes: temperate forests and boreal forests, following the classification of the World Wildlife Foundation (http://worldwildlife.org)[73].

## Statistical analyses

The C accumulation in tree biomass, soil organic and mineral horizons, and the N accumulation in soil organic and mineral horizons were considered our response variables and analyzed separately. To explore how the relationships between tree functional diversity and C and N accumulation in trees and soils varied across environmental resource gradients after accounting for the effect of stand age and tree community functional identity, we used the following linear model:

$$
\begin{aligned}
\triangle C_{Tree}, \triangle C_{Organic}, \triangle C_{Mineral}, \triangle N_{Organic} \text{ or} \triangle N_{Mineral} =& \beta_0 + \beta_1 \cdot SA + \beta_2 \cdot Solar \\
&+ \beta_3 \cdot CMI + \beta_4 \cdot ND + \beta_5 \cdot FD_{is} + \beta_6 \cdot CWM_{PC1} + \beta_7 \cdot CWM_{PC2} \\
&+ \beta_8 \cdot Solar \times FD_{is} + \beta_9 \cdot Solar \times CWM_{PC1} + \beta_{10} \cdot Solar \times CWM_{PC2} \quad (1)\\
&+ \beta_{11} \cdot CMI \times FD_{is} + \beta_{12} \cdot CMI \times CWM_{PC1} + \beta_{13} \cdot CMI \times CWM_{PC2} \\
&+ \beta_{14} \cdot ND \times FD_{is} + \beta_{15} \cdot ND \times CWM_{PC1} + \beta_{16} \cdot ND \times CWM_{PC2} + \varepsilon
\end{aligned}
$$

where $\beta_i$ and $\varepsilon$ are the coefficients to be estimated and the sampling error, respectively. $FD_{is}$, $CWM_{PC1}$, and $CWM_{PC2}$ are functional diversity and CWM of the first and second dimensions of variations in functional traits. The Solar, CMI, and ND are environmental gradient indices for solar energy, water availability, and N availability. The stand age (SA) and forest functional identity ($CWM_{PC1}$ and $CWM_{PC2}$) were included as covariates. As an alternative to ND, we also investigated how the relationships between tree functional diversity and C and N accumulation in trees and soils varied across a soil N availability gradient by replacing ND with soil C/N ratios in the organic or mineral horizon.

To explore how the relationships between tree functional diversity and C and N accumulation in trees and soils varied across environmental non-resource stress factors after accounting for the effect of stand age and tree community functional identity, we used the following linear model:

$$
\begin{aligned}
\triangle C_{Tree}, \triangle C_{Organic}, \triangle C_{Mineral}, \triangle N_{Organic} \text{ or} \triangle N_{Mineral} =& \beta_0 + \beta_1 \cdot SA \\
&+ \beta_2 \cdot HI/pH + \beta_3 \cdot FD_{is} + \beta_4 \cdot CWM_{PC1} + \beta_5 \cdot CWM_{PC2} \\
&+ \beta_6 \cdot HI/pH \times FD_{is} + \beta_7 \cdot HI/pH \times CWM_{PC1} \quad (2)\\
&+ \beta_8 \cdot HI/pH \times CWM_{PC2} + \varepsilon
\end{aligned}
$$

where HI and pH are non-resource stress indices for heatwave and soil acidity. The stand age (SA) and forest functional identity ($CWM_{PC1}$ and $CWM_{PC2}$) were included as covariates.

All explanatory variables were centered and scaled (mean = 0, SD = 1) before the analysis to permit the proper comparison of the resulting model coefficients. To prevent overfitting[74], we selected the most parsimonious model based on the lowest AIC among all alternatives using the 'dredge' function in the muMIn package[75] (Fig. 2, Supplementary Tables 1-4). Collinearity among explanatory variables was tested by variance inflation factors (VIFs). We found that all predictors had VIF < 5; hence, there was no multicollinearity in the most parsimonious models[76]. Thus, we report results for the most

parsimonious models in this paper. However, solar radiation is highly geographically intercorrelated with MAT (Supplementary Figs. 1, 2), which is a surrogate for chemical energy and plays a dominant role in controlling ecological systems and processes. We used a stepwise approach to disentangle the impacts of solar radiation from temperature. First, we replaced the term solar radiation with MAT in the initial full models and then included both terms in the initial full models, yet we did not find any significant interactive effects between MAT and $FD_{is}$ on C and N accumulation in trees and soils. Second, we added MAT and its interaction with $FD_{is}$ as a covariate in the most parsimonious model and found that the models with and without MAT and its interaction yielded qualitatively similar estimates and trends (Supplementary Table 8). In the model including both solar radiation and MAT on tree biomass C accumulation, only solar radiation was significant (F = 10.84, df = 1, P = 0.001, Supplementary Table 8). The linear three-dimensional regression model that included both solar radiation and MAT also showed that C accumulation in tree biomass depended not only on MAT, but also on solar radiation (Supplementary Fig. 5c). Finally, given that solar radiation was included in the most parsimonious model for C accumulation in tree biomass, we calculated the residuals of C accumulation in tree biomass with respect to MAT and subsequently modeled these residuals instead of raw data of tree biomass C accumulation as a function of variables in the most parsimonious model to avoid problems stemming from correlations and colinearity between solar radiation and MAT. Even after the MAT effect was removed, the solar radiation effect on tree biomass C accumulation was still significant (F = 6.64, df = 1, P = 0.010, Supplementary Fig. 5b). Even after all plots were grouped into two biomes (temperate and boreal forests) based on the mean annual air temperature, we did not found significant interactions between $FD_{is}$ and biome on C and N accumulation in trees and soils (Supplementary Table 9).

We used partial regressions to graphically illustrate the effects of Solar, CMI, ND, HI, and soil pH on relationships between tree functional diversity and C and N accumulation in trees and soils. We calculated Solar-, CMI-, ND-, HI- and soil pH-dependent functional diversity effects as $\beta_0 + \beta_5 \cdot FD_{is} + \beta_2 \cdot Solar$ (or CMI, ND, HI, pH) $\times FD_{is}$ for the mean, and mean plus and minus one standard deviation of Solar, CMI, ND, HI, and soil pH, respectively. We examined the spatial dependence of model residuals using Moran's I test implemented in the spdep package in R[77], revealing no discernible autocorrelation. Significance was set at $\alpha = 0.05$ for all analyses unless otherwise stated. We used a one-sided F-test to calculate P values. All statistical analyses were performed in R 4.3.1[78].

## Reporting summary

Further information on research design is available in the Nature Portfolio Reporting Summary linked to this article.

## Data availability

The data on tree biomass carbon accumulation, soil carbon and nitrogen accumulation, and local environmental condition generated in this study have been deposited in the Figshare database at https://doi.org/10.6084/m9.figshare.25037213. The raw tree and soil data are available under restricted access for scientific research-only use, and access can be obtained by application at https://nfi.nfis.org/en/.

## Code availability

The code used in this study is available in the Figshare database at https://doi.org/10.6084/m9.figshare.25037213.

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

## Acknowledgements

We are grateful to E. B. Searle for his editorial comments during the revision. Canadian Forest Service of the Natural Resources Canada shared the data from the National Forest Inventory database. X.c. acknowledges the support from National Natural Science Foundation of China (NSFC, 32401546), NSFC Excellent Young Scientists Fund (overseas), the Scientific Research Startup Fund Project of Zhejiang A&F University (2024LFR019), NSERC in the form of a and a MITACS internship. S.X.C. acknowledges the support from a Discovery grant (RGPIN-2018-05700) of the Natural Sciences and Engineering Research Council of Canada (NSERC). PBR was supported by the U.S. National Science Foundation ASCEND Biology Integration Institute (NSF-DBI-2021898).

## Author contributions

X.C., P.B.R., and S.X.C. were responsible for the conception and design of the project. X.C. and A.R.T. compiled data. X.C. analyzed the data and wrote the first draft of the manuscript. X.C., P.B.R., A.R.T., Z.A., and S.X.C. contributed to reviewing and editing. S.X.C. supervised the work and acquired funding. All authors approved the final manuscript.

## Competing interests

The authors declare no competing interests.

## Inclusion & ethics

For this research, local researchers were included throughout the research process including study design, study implementation, data ownership, and authorship. Contributors who do not meet all criteria for authorship have been listed in the Acknowledgements section. All roles and responsibilities were agreed amongst collaborators ahead of the research. We have considered local and regional research relevant to our study in the citations. This study does not involve human research participants or animals and does not require approval by a local ethics review committee.
