## [Peer Review File · Nature Communications]

Resource availability enhances positive tree functional diversity effects on carbon and nitrogen accrual in natural forestsEditorial Note: This manuscript has been previously reviewed at another journal that is not operating a transparent peer review scheme. This document only contains reviewer comments and rebuttal letters for versions considered at Nature Communications.

REVIEWER COMMENTS

Reviewer #2 (Remarks to the Author):

Thank you for the authors to revise the manuscript according to my last comments, especially showing more details about the statistics and results. However, there are still major concerns to me:

1) The concept figure is a big confusing: Figure 1b is already the results from the study (as it says "in the most parsimonious model"? Or is it only the hypothesis? If it is the former, it should not appear in Figure 1, but more like a summary appearing in the end as a result; If it is the later, then the authors need to give the reason why predicted as it shows.

2) I was wonder why need to exclude "tree identity" if all the analysis conducted at the plot level? Do you mean plot composition? (see line 417, 434 et al.)

3) There is no reason why should "forest type" included in the analysis based on the authors' concept. There is nothing related in the introduction and no discussion about the results.

4) I am trying hardly to match the figures with the supplenetary tables and the results. The author strongly concludes "strong evidence that the functional diversity-induced increases of C and N accumulation in both tree biomass and soils were stronger in solar radiation-, water- and nutrient-rich environments" (line 200-202). However, the results are bit mixed. For tree biomass, it is true in solar radiation- and nutrient-rich environments. While for soil carbon, it is more related to nutrient and heatwave intensity, rather than water. The author has to carefully give the conclusion.

4) In the method, there are too many discussion about the results. e.g. line 446-452 should not be in the method part. There are more like such.

Reviewer #4 (Remarks to the Author):

Dear authors, after reading every response and revision made in the manuscript (for both reviewers 1 and 3) I consider that these are appropriate and that you have made a great effort and work to adapt the comments provided by both reviewers to the latest version of the manuscript. I consider that the manuscript is well written and, although it does not provide very novel results, it may have an interesting contribution, mainly related to the incorporation of temporal information of soil variables to data with important spatial amplitude in the temperate-boreal climate transition. I would only add that you should also tone down the claim that functional diversity increases carbon accumulation in a unidirectional causal sense. It could be that in more productive sites, this causal relationship is reversed and it is higher productivity that generates higher diversity according to the diversity-energy hypothesis. In case there is not pure experimental information, causality could change along productivity gradients in forests ecosystems (there are a few papers on this since 2020).

Reviewer #4 (Remarks on code availability):

I have checked the code and believe that everything is OK. Otherwise, I did not have the time to make an in detail review of the code.

I couldn't see a README file in the code nor specific explanation on the lines of the code.

Resource availability enhances positive tree diversity effects on carbon and nitrogen accrual in natural forests

Responses to Reviewers' Comments:

Reviewer #2 (Remarks to the Author):

Thank you for the authors to revise the manuscript according to my last comments, especially showing more details about the statistics and results. However, there are still major concerns to me:

Response: We greatly appreciate that you took the time to review our manuscript so carefully. We are pleased with your positive evaluation of our work and with your constructive suggestions. We have carefully considered each of your comments and responded to them and have revised the manuscript accordingly.

1) The concept figure is a big confusing: Figure 1b is already the results from the study (as it says “in the most parsimonious model”? Or is it only the hypothesis? If it is the former, it should not appear in Figure 1, but more like a summary appearing in the end as a result; If it is the later, then the authors need to give the reason why predicted as it shows.

Response: Thank you for your insightful comments. We do apologize if we didn't fully address the reviewer's comment in the first round of revision before submitting to Nature Communications. To address the comment, we have separated Figure 1a and Figure 1b into two distinct figures. The revised **Figure 1** now exclusively presents the conceptual diagram of environmental context-dependent effects, serving to present our hypotheses. The former Figure 1b, which depicts the results from the most parsimonious model, has been moved to **Extended Data Figure 1**. This distinction should clarify the purpose of each figure: Figure 1 as the hypothesis and Extended Data Figure 1 as the resulting outcomes.

2) I was wonder why need to exclude “tree identity” if all the analysis conducted at the plot level? Do you mean plot composition? (see line 417, 434 et al.)

Response: We are sorry for the lack of clarity. We have revised it as “community functional identity” (**Line 402 and 420**)

3) There is no reason why should “forest type” included in the analysis based on the authors' concept. There is nothing related in the introduction and no discussion about the results.

Response: Thank you for your insightful comments. We have removed all text about forest types (previous **Supplementary Tables. 5 & 6** and **Supplementary Figs. 5 & 6** and the corresponding results)

4) I am trying hardly to match the figures with the supplementary tables and the results. The author strongly concludes “strong evidence that the functional diversity-induced increases of C and N accumulation in both tree biomass and soils were stronger in solar radiation-, water- and nutrient-rich environments” (line 200-202). However, the results are bit mixed. For tree biomass, it is true in solar radiation- and nutrient-rich environments. While for soil carbon, it is more related to nutrient and heatwave intensity, rather than water. The author has to carefully give the conclusion.

Response: We have revised the manuscript to make our writing clearer. (Line 200-205). The revised sentence is:

“Specifically, we found that the positive association between tree functional diversity and C accumulation in tree biomass was more pronounced in solar radiation-, and nutrient-rich environments. Additionally, the positive relationship between functional diversity and C and N accumulation in the organic soil horizon was more pronounced in high-water and low-heatwave conditions, while in mineral soil horizons, these relationships were stronger in nutrient-rich and high-heatwave environments.”

We separated the sentences for trees and soils to provide more detailed explanations for each.

4) In the method, there are too many discussion about the results. e.g. line 446-452 should not be in the method part. There are more like such.

Response: We have moved previous “line 446-452” to the results section (Line 136-143). For the remaining discussion in the Methods section (Lines 438-457), we believe it is appropriate to retain it there, as it serves to demonstrate the robustness of our methodology rather than presenting our main results. However, we have provided a brief introduction and discussion in the main text (Lines 124-127, 145-151), while leaving the detailed explanation in the Methods section..

Reviewer #4 (Remarks to the Author):

Dear authors, after reading every response and revision made in the manuscript (for both reviewers 1 and 3) I consider that these are appropriate and that you have made a great effort and work to adapt the comments provided by both reviewers to the latest version of the manuscript. I consider that the manuscript is well written and, although it does not provide very novel results, it may have an interesting contribution, mainly related to the incorporation of temporal information of soil variables to data with important spatial amplitude in the temperate-boreal climate transition. I would only add that you should also tone down the claim that functional diversity increases carbon accumulation in a unidirectional causal sense. It could be that in more productive sites, this causal relationship is reversed and it is higher productivity that generates higher diversity according to the diversity-energy hypothesis. In case there is not pure experimental information, causality could change along productivity gradients in forests ecosystems (there are a few papers on this since 2020).

Response: We greatly appreciate your constructive comments. We have toned down our claim that functional diversity increases carbon accumulation in a unidirectional causal way (Line 29-30, 170-172, 180, 200-205, 239, 241, 257, 283, 285, 287)

Reviewer #4 (Remarks on code availability):

I have checked the code and believe that everything is OK. Otherwise, I did not have the time to make an in detail review of the code.

I couldn't see a README file in the code nor specific explanation on the lines of the code.

Response: Thank you for reviewing the code. We have re-visited the code and have added an explanation for the lines to make the codes easier to follow.